# Evaluating Pediatric NAFLD with Controlled Attenuation Parameter: A Comprehensive Narrative Review

**DOI:** 10.3390/diagnostics15030299

**Published:** 2025-01-27

**Authors:** Ingrid Arteaga, Carla Chacón, Alba Martínez-Escudé, Irene Ruiz Rojano, Galadriel Diez-Fadrique, Meritxell Carmona-Cervelló, Pere Torán-Monserrat

**Affiliations:** 1Unitat de Suport a la Recerca (USR) Metropolitana Nord, Fundació Institut Universitari d’Investigació en Atenció Primària Jordi Gol i Gurina (IDIAP Jordi Gol), 08303 Mataró, Spain; iarteaga@gencat.cat (I.A.); cchaconv.mn.ics@gencat.cat (C.C.); amescude.mn.ics@gencat.cat (A.M.-E.); ireneruiz.mn.ics@gencat.cat (I.R.R.); galadrieldiez21@gmail.com (G.D.-F.); mcarmonace.mn.ics@gencat.cat (M.C.-C.); 2Grup de Recerca en Malalties Hepàtiques a l’Atenció Primària (GRemHAp), IDIAP Jordi Gol, USR Metro-Nord, 08303 Mataró, Spain; 3Primary Healthcare Center Vall del Tenes, Gerència d’Àmbit d’Atenció Primària Metropolitana Nord, Institut Català de la Salut, 08186 Llicà d’Amunt, Spain; 4Ph.D. Programme in Medicine and Translational Research, Faculty of Medicine, University of Barcelona, 08193 Barcelona, Spain; 5Primary Healthcare Center La Llagosta, Gerència d’Àmbit d’Atenció Primària Metropolitana Nord, Institut Català de la Salut, 08120 La Llagosta, Spain; 6Department of Medicine, Faculty of Medicine, Autonomous University of Barcelona, 08193 Bellaterra, Spain; 7Primary Healthcare Center Dr. Barraquer, Gerència d’Àmbit d’Atenció Primària Metropolitana Nord, Institut Català de la Salut, 08930 Sant Adrià del Besos, Spain; 8Germans Trias i Pujol Research Institute (IGTP), 08916 Badalona, Spain

**Keywords:** NAFLD, non-alcoholic fatty liver disease, CAP, controlled attenuation parameter, elastography, liver, children, adolescents, pediatrics

## Abstract

Non-alcoholic fatty liver disease (NAFLD) in the pediatric population has emerged as a significant health concern due to its alarming rise in prevalence. In children, the characteristics of the disease differ from those seen in adults. NAFLD may progress to more severe liver disease in children compared to adults with similar profiles. Liver biopsy remains the gold standard for diagnosis; its invasive nature and high cost limit its use as a first-line tool. Alternatively, magnetic resonance imaging (MRI) techniques, such as magnetic resonance imaging-estimated liver proton density fat fraction (MRI-PDFF), have shown a good correlation with the degree of histological steatosis, although their use is limited by high costs and limited accessibility. Controlled attenuation parameter (CAP), integrated with vibration-controlled transient elastography (VCTE) (FibroScan^®^), is a novel non-invasive, accessible, and effective method for diagnosing hepatic steatosis. In this article, we reviewed the existing literature on the diagnostic accuracy of CAP in pediatric NAFLD. The PubMed and EMBASE databases were searched. Seven relevant studies were identified, conducted in pediatric hospital populations with specific demographic characteristics. Two of these studies compared CAP with liver biopsy, one compared CAP with liver biopsy and MRI-PDFF, and the remaining four compared CAP with MRI. Overall, CAP proved to be accurate in detecting the presence or absence of fatty infiltration, positioning it as a promising tool to simplify the diagnosis of NAFLD in children. However, further studies in larger populations are needed to confirm these findings and facilitate its implementation in routine clinical practice.

## 1. Introduction

### 1.1. Understanding NAFLD in Pediatric Populations

The first cases of pediatric NAFLD were reported in 1983 in three patients with obesity [1]. Since then, over four decades later, NAFLD has emerged as the most prevalent chronic liver disease in children and adolescents, with prevalence rates ranging from 7.6% in the general pediatric population to 34.2% among those with obesity [2,3]. The pathogenesis of NAFLD involves both genetic predispositions and environmental factors, particularly sedentary behavior and high-calorie diets [4]. Adolescence represents a critical period when hormonal changes and other factors contribute to an increased disease prevalence [5]. Males are disproportionately affected by NAFLD, and some evidence suggests the protective role of estrogens in females [3,6]. Ethnicity also plays a key role in disease prevalence, with studies reporting the highest rates of NAFLD in the Hispanic population, followed by the Caucasian and African-American groups [3,7].

Pediatric NAFLD is defined as the presence of chronic hepatic steatosis in children under 18 years of age, in whom it has been ruled out as secondary to genetic/metabolic disorders, infections, the use of steatogenic drugs, alcohol consumption, or malnutrition [8]. NAFLD is mostly associated with insulin resistance (IR), central or generalized obesity, and dyslipidemia [8,9]. Recently, a new nomenclature has been proposed named metabolic dysfunction-associated steatotic liver disease (MASLD), emphasizing, among other factors, its association with metabolic dysfunction [10]. Furthermore, a position paper endorsed by several scientific societies supports the adoption of this new terminology for pediatric hepatic steatosis disease [11]. This endorsement underscores the importance of considering diagnostic criteria and exploring alternative diagnoses, emphasizing the differences in relation to the adult population. Likewise, they refer to the need for implementing practical applications, conducting further research, advocating for awareness, and enhancing pediatrician education in this area [11].

Children with NAFLD may exhibit similar histologic patterns to those seen in adults, with inflammation, steatosis, and perisinusoidal fibrosis in zone 3 [12]. Adolescents are more likely to exhibit the classic steatohepatitis (NASH) pattern commonly observed in adults. In contrast, preadolescent children tend to show an alternative pattern of liver injury, characterized by panacinar steatosis, portal inflammation, and/or periportal fibrosis, predominantly affecting zone 1 [3,12,13]. Some authors suggest that puberty influences the pathological features of the disease, with less severe manifestations during and after puberty [5]. Post pubertal individuals are less likely to exhibit zone 1 steatohepatitis but are more prone to having Mallory–Denk bodies [5,12]. Additionally, a high prevalence of metabolic comorbidities, such as IR, dyslipidemia, and metabolic syndrome (MetS), has been seen in adolescents with severe obesity. When analyzing liver biopsies from these adolescents and comparing them with a group of adults with similar characteristics (undergoing bariatric surgery), it was found that adolescents had a higher incidence of steatohepatitis and liver fibrosis at more advanced stages than adults [14]. These findings suggest a more aggressive progression of the disease in younger individuals. In this context, some authors reported a concerning figure: 17% of children with overweight and obesity, referred from primary care to specialized care and diagnosed with NAFLD by liver biopsy, had advanced fibrosis (F3) at the time of diagnosis [15]. Despite this, it is worth noting that NAFLD can be found in five to eight percent of children who have a normal weight [3,16].

Although these data highlight the importance of diagnosing the disease, it is important to remember that most children with NAFLD are usually asymptomatic. However, symptoms such as fatigue, irritability, difficulty concentrating, pain, or muscle cramps have been reported as accompanying manifestations [17]. Additionally, there are comorbidities that have been described that are associated with an increased risk of NAFLD, such as obstructive sleep apnea, diabetes mellitus type 2 (DM2), panhypopituitarism, and polycystic ovary syndrome [18]. It is well established that NAFLD in the adult population shares several cardiovascular disease risk factors that are also components of MetS [19,20]. In children, some authors emphasize the association between MetS and NAFLD [21,22,23]. Furthermore, it has been observed that the presence of overweight and obesity during adolescence is strongly linked to an increased risk of cardiovascular mortality in adulthood [24]. In this context, a retrospective analysis of 210 patients aged 0 to 18 years with NAFLD and referred to a specialized liver unit in Singapore between 2003 and 2020 revealed several concerning trends: an increase in NAFLD referrals, a rise in MetS-associated comorbidities at younger ages, and the potential risk of advancing to late adolescence with advanced fibrosis [25].

Nevertheless, the natural history of pediatric NAFLD is still controversial. There have been only a few reported cases in which cirrhosis developed in young children as a result of NAFLD [13,26]. To date, the only long-term study, with a 20-year follow-up, was conducted in obese children evaluated at a tertiary hospital referral center [27]. This study found that NAFLD in these children was more likely to progress to more severe liver disease and had a higher mortality rate compared to healthy controls. The researchers observed that four out of five biopsies that initially did not show fibrosis developed fibrosis within a 3.4-year follow-up period. On the other hand, in two double-blind, randomized clinical trials that compared paired liver biopsies from 122 of 139 children with NAFLD who received a placebo and standard care lifestyle advice, one-third of the children in the placebo groups showed histologic progression within two years [28]. This suggests that fibrosis progression in children might be slow.

Pediatric NAFLD has distinctive characteristics compared to those of adults and its diagnosis requires the careful exclusion of alternative conditions. This assessment must take into account demographic, anthropometric, clinical, and laboratory features [8,29,30,31].

### 1.2. Assessment of Steatosis in Children and Adolescents

Steatosis is the result of the accumulation of triacylglycerol in greater than five percent of liver cells and can have different etiologies (alcohol consumption, viral hepatitis, or metabolic dysfunction) [9,32]. In the new nomenclature, the term “steatotic liver disease” (SLD) serves as the umbrella under which disorders caused by hepatic fat accumulation are grouped. Pediatric liver steatotic disease without warning signs, in the presence of cardiometabolic criteria and after ruling out other primary causes, leads us to diagnose the presence of MASLD [10,11]. In pediatrics, an active search for the primary causes of SLD should be conducted, including autoimmune hepatitis, Wilson’s disease, viral hepatitis, alpha-1 antitrypsin deficiency, celiac disease, and the detection of any alcohol consumption in older adolescents, as well as potential drug-induced liver injury (DILI) [11,33]. The process of hepatocyte fat accumulation is intricate, involving multiple factors and a pathophysiological mechanism. Although the two-hit theory was initially proposed, current evidence suggests a multi-hit process involving a variety of overlapping pathways [34]. While typically reversible and benign, this fat accumulation can, in many instances, coincide with inflammation, leading to NASH [35]. NASH is a dynamic process that can regress to steatosis [36]. However, persistent inflammation over the years may trigger fibrosis, which in turn may in some patients trigger and lead to cirrhosis, liver failure, or hepatocellular carcinoma [31]. Research observed that patients with NAFLD who do not have features of NASH are still at some risk of fibrosis progression [37,38]. In fact, children with the pediatric pattern of NAFLD can develop advanced fibrosis without exhibiting the typical features of NASH [12].

Liver biopsy remains the gold standard for diagnosis due to its superior accuracy in identifying steatohepatitis and advanced fibrosis compared to other methods, such as imaging or serological tests. However, non-invasive tests provide a valuable alternative to liver biopsy. Although they do not match the diagnostic precision of biopsy, the combined use of various non-invasive diagnostic methods can significantly enhance diagnostic accuracy [39].

Current guidelines stress the importance of excluding other causes of steatosis, particularly in younger patients, and using liver biopsy exclusively to resolve diagnostic doubts [8,29,30,31]. This is reasonable, considering the impact that an invasive procedure, which is not without side effects, can have on both the patient and their parents [40]. Moreover, there is heterogeneity in the recommendations of different scientific societies about screening for the disease, including differences in age, clinical characteristics, and risk factors. Some guidelines propose starting screening between 9 and 11 years of age in the presence of overweight, obesity, or other risk factors [8], while others, due to a paucity of evidence, do not formally recommend screening in children [31]. In response to this scenario, non-invasive diagnostic tools have been developed, including serological markers, imaging, and elastographic tests, which can contribute to the non-invasive diagnosis of the disease.

#### 1.2.1. Blood Test and Serological Markers

Alanine aminotransferase (ALT) levels are likely the most widely used biomarker in clinical practice for assessing pediatric NAFLD. ALT values > 80 U/L can be used as a non-invasive test for the diagnosis of NAFLD [8,41]. Nonetheless, some children with NAFLD may present with ALT values within the normal range, leading to underdiagnosis when ALT is the sole diagnostic tool, especially in the obese population, in which the prevalence of NAFLD appears lower if based on ALT alone [2]. Although several serological markers have been validated for diagnosing NAFLD in adults, such as the Fatty Liver Index (FLI) for steatosis and the NAFLD Fibrosis Score or Fibrosis-4 (FIB-4) score for fibrosis, these markers have not demonstrated sufficient accuracy for pediatric NAFLD diagnosis when compared to liver biopsy [42,43,44,45]. With this in mind, validation studies for its use in pediatric populations are needed.

#### 1.2.2. Imaging Diagnosis

There are many tools for NAFLD diagnosis using imaging techniques: ultrasonography (US), computed tomography (CT), and magnetic resonance imaging (MRI). In the adult population, hepatic ultrasound exhibits a high accuracy in differentiating moderate-to-severe steatosis from individuals without this condition, using liver biopsy as the reference standard. It achieves a sensitivity of 85% and a specificity of 94% [46]. In pediatric imaging diagnostics, there is no uniform consensus on recommending ultrasound as a first-line test. Ultrasound has a positive predictive value (PPV) between 47% and 62% for identifying steatosis in children [47]. However, limited sensitivity and specificity levels have been reported for detecting steatosis, particularly in cases of mild steatosis (S0–S1), in which fatty infiltration is less than 33% [48]. In this regard, scientific societies advocate for its use in high-risk patients, acknowledging its value in clarifying the diagnosis of NAFLD [29,30], whereas others do not propose it as a primary tool in children [8]. Recently, new tools for diagnosis of steatosis like the shear wave elastography (SWE) and acoustic radiation force impulse (ARFI) methods have been development to allow for the quantitative evaluation of fibrosis on sonograms [49].

In recent years, quantitative magnetic resonance imaging, specifically MRI-estimated liver proton density fat fraction (PDFF), has shown a higher and better diagnostic accuracy than ultrasound [50]. MRI-PDFF decomposes the liver signal into water and fat, subsequently determining the fractional fat content of the liver, which serves as an indicator of hepatic triglyceride concentration. Several authors report a strong correlation with the histological grade [51,52,53]. Although it has a good correlation with the degree of steatosis and a high positive predictive value, its high cost and limited accessibility in most clinics prevent it from being used as a first-line test [54,55,56,57].

#### 1.2.3. Controlled Attenuation Parameter

The evaluation of steatosis using CAP is performed simultaneously with VCTE, using both the same radiofrequency and anatomical site to assess liver stiffness. Elastography tests to diagnose liver parenchymal stiffness and enable the diagnosis of fibrosis can be performed by different methods, either alone by VCTE or as elastography techniques integrated with ultrasound or magnetic resonance devices. VCTE, using the Fibroscan^®^ device, has appeared as a useful and simple tool to diagnose liver stiffness and detect liver fibrosis in adults and has been widely validated [58]. CAP is an algorithm that has been incorporated into the Fibroscan^®^. The term CAP derives from its specific targeting of liver tissue and has been developed to offer immediacy, reproducibility, and independence from both operators and machines [59]. It is a simple device that assesses the degree of ultrasound beam attenuation resulting from the mechanical impulse of the Fibroscan^®^ probe in the hepatic parenchyma. This mechanical impulse generates an acoustic wave that passes through the liver parenchyma. The higher the fat content, the greater the attenuation of the ultrasound wave. Ultrasound attenuation will correspond to the loss of ultrasound energy as it traverses the liver. Thus, the attenuation coefficient is an estimate of the ultrasound attenuation for a given frequency depending on the probe used. For the M probe, the frequency is 3.5 MHz, and the XL probe uses a lower central frequency (2.5 MHz) and measures deeper below the skin surface (3.5–7.5 cm). According to the manufacturer, the M probe should be used when the skin–liver capsule distance (SCD) is <25 mm, whereas the XL probe should be used when the SCD is ≥25 mm. The attenuation coefficient is expressed in dB/m and can range from 100 to 400 dB/m [59]. An interquartile range (IQR) has been proposed to assess the quality of the measurements (IQR < 30 or 40 dB/m) [60,61]. Evidence of the utility of CAP as a very good tool for diagnosing steatosis in adults has been widely demonstrated [62]. CAP has demonstrated high sensitivity and specificity levels in detecting mild steatosis when compared to liver biopsy [63]. In the adult population, the cut-off point for S1 ranges between 247 and 302 dB/m, and values above >275 dB/m have a sensitivity and a positive predictive value (PPV) greater than 90% for detecting steatosis, while values ≥288 dB/m show a sensitivity of 75% and a PPV of 90% for detecting steatosis >5% [60,64]. Nevertheless, despite growing evidence of its applicability in the pediatric and adolescent population, the usefulness of CAP as a diagnostic tool for steatosis in these age groups remains limited [65,66,67]. Scarce studies have been published to date, with most of them having a small number of participants and being conducted in specific population groups. In addition, few studies have compared CAP with liver biopsy as the gold standard test, because of its invasiveness, and have used other non-invasive tests as a reference [40]. Consequently, the use of non-invasive tests for NAFLD diagnosis results in heterogeneous recommendations across various clinical practice guidelines [8,29,30,31].

The longevity of children, controversial data on the natural history of the disease in pediatrics, and the increase in its prevalence make it essential to have tools that simplify diagnosis and that are easy, innocuous and, above all, accessible in most clinics. Thus, CAP has positioned itself as a simple, non-invasive, and reproducible alternative for the diagnosis of steatosis. The overall objective of this narrative review was to provide an overview of the advancements in CAP’s diagnostic performance for detecting NAFLD in the pediatric and juvenile population, drawing from the available literature.

## 2. Materials and Methods

Although MASLD has recently been proposed as a new nomenclature, we have compiled scientific literature on the diagnostic accuracy of CAP in pediatric NAFLD, as this was the predominant terminology in the literature prior to this review [10,11]. A literature search was conducted on the electronic databases of PUBMED and EMBASE. Eligibility was restricted to studies published in English and conducted between 2014 and 2023. The following combination of terms was used: (*Non-alcoholic fatty liver disease*) AND (*pediatric*) AND (*controlled attenuated parameter*). Once the search was completed, duplicated records were eliminated, and 84 articles were obtained. The remaining articles were then screened based on their titles and abstracts. The abstracts of each article were reviewed applying the following inclusion criteria: studies designed to evaluate NAFLD in the pediatric population (children and adolescents), using CAP as the diagnostic method and liver biopsy and/or MRI as the reference method. As we mentioned previously, liver biopsy is the gold standard for diagnosing the disease; however, the inherent limitations of the test contribute to the limited literature available using this diagnostic method in the pediatric population. We chose to include MRI because it is becoming more widely accepted as a substitute for liver biopsy, especially in the pediatric population [56]. We excluded studies that used ALT, ultrasound, or non-invasive serological methods as the reference method for assessing steatosis compared to CAP. Studies that passed this initial screening were evaluated using the full-text versions to ensure that all predefined eligibility criteria were met. From the selected studies, relevant information was extracted, and the data obtained were synthesized to provide an overview of the current state of research on the performance of CAP for pediatric NAFLD diagnostics.

## 3. Results

Of the 84 studies obtained, seven met the established eligibility criteria, while 77 were excluded for various reasons, including having objectives different from those set for this review, specifically: including a particular adult population, not making comparisons to the gold standard, and being of other article types such as study protocols or clinical guidelines, among others.

### 3.1. Assessment of CAP Diagnostic Accuracy in Contrast to Liver Biopsy

Studies comparing CAP with liver biopsy as the reference standard, published up to the time of writing this review, are listed in Table 1.

In a cross sectional study, Desai et al. [68] included 149 participants who had undergone a liver biopsy as part of routine clinical care and had a variety of liver pathologies, and CAP assessment was performed within one year of biopsy (mean, 1.3 months; IQR, 0.5–3.2). CAP measurements were determined using the M (*n* = 62) or XL (*n* = 5) probe. Participants with S probe measurements and those with invalid measurements were excluded (*n* = 76). Of the 69 participants included, 84% were under 18 years of age and 62% were male. Twenty-three participants had steatosis, with 14 having NAFLD, of whom 10 had NASH. The CAP values were significantly higher in subjects with steatosis, regardless of whether they were normal weight, overweight, or obese. The mean BMI of those with steatosis was higher than that of those without steatosis. When analyzing the effect of overweight and obesity on CAP measurements, as well as the interaction between weight status (overweight/obesity: yes/no) and the presence of steatosis (yes/no), no significant interaction was found (*p* = 0.81). The mean CAP measurement in subjects without steatosis was lower than that in subjects with steatosis (198 ± 37 dB/m vs. 290 ± 47 dB/m, *p* < 0.0001). The results showed that CAP could differentiate between steatosis severity, with values of 265 ± 53 dB/m for mild or moderate steatosis and 313 ± 25 dB/m for those with marked steatosis (*p* < 0.001). An optimal cut-off point of 225 dB/m was identified to detect fatty infiltration values above five percent, as determined by liver biopsy, with sensitivity and specificity levels greater than 80%.

Chaidez et al. [69] conducted a study to determine whether LSM and CAP correlated with the histological stage of liver fibrosis and the degree of steatosis, respectively. The predictive ability of the FibroScan-AST (FAST) score in pediatric NAFLD was also analyzed. The study included 68 participants in the prospective cohort, who already had a scheduled clinical liver biopsy, and 147 in the retrospective cohort, based on a review of the clinical FibroScans obtained as part of standard care. Nine participants were excluded (three for previous liver transplantation and six for other reasons), resulting in a total of 206 participants (61% male; 53% Hispanic). The participants were grouped into two cohorts: 106 with NAFLD and 90 without NAFLD. Significant correlations were found between LSM and fibrosis stages, with the strongest correlation observed in the non-NAFLD group (Spearman’s r = 0.47, *p* < 0.0001). In the NAFLD group, the majority had grade 1 to 3 steatosis, while in the non-NAFLD group, grade 0 steatosis predominated. In addition, participants with NAFLD had significantly higher mean BMI scores and CAP values than the non-NAFLD group. The mean CAP value in those with steatosis grade 0 (S0) was 201 ± 42 dB/m. The CAP showed an AUROC of 0.98 to differentiate steatosis grades 1 to 3 from grade 0, with a strong positive correlation between CAP values and histological steatosis grade. A cut-off point ≥259 dB/m predicted steatosis grades 1 to 3 with sensitivity and specificity levels greater than 90%. A significant positive correlation was also observed between BMI and CAP values (*p* < 0.0001).

Finally, Alves et al. [70] conducted a prospective study to evaluate the technical performance of VCTE in pediatric NAFLD. The study determined the concordance between VCTE and reference standards, such as MR elastography, MRI-PDFF, and liver biopsy, in a total of 84 participants (65% male). One participant was excluded due to a chest circumference ≤75 cm and three others because their examinations were incomplete. In the end, valid CAP measurements were obtained in 80 of the 83 participants, of whom 64 were under 18 years of age. A total of 57% of the participants were tested with the M probe and 94% were obese. The mean CAP value was 314 ± 50 dB/m, and a CAP value > 249 dB/m was considered indicative of steatosis grade ≥ 1 for VCTE performance analysis. To maximize measurement stability, steatosis-related measurements were recorded only if they were performed within 3 months of the transient elastography examination. Sixteen MRI scans and six liver biopsies were used for steatosis comparisons. No significant correlation was found between CAP values and the degree of liver steatosis determined by biopsy. The authors attribute this lack of correlation to the absence of a non-NAFLD control group and the small sample size.

### 3.2. Assessment of CAP Diagnostic Accuracy in Contrast to Imaging Tests

Four studies were identified that compared MRI-PDFF with CAP. One of these studies also used liver biopsy, as previously described. In addition, one study comparing liver ultrasound, using proton density magnetic resonance spectroscopy of the fat fraction with proton density (MRS-PDFF) as a reference, was included. The studies are summarized in Table 2.

All participants included in these studies were recruited from hospitals or specialized obesity clinics [70,71,72,73,74]. The first of these studies, conducted in South Korea by Shin et al. [71], involved a retrospective cohort of 86 participants with a mean age of 13.1 ± 2.7 years. Participants underwent MRI-PDFF and CAP measurements to assess NAFLD as part of routine clinical practice. CAP measurements were performed within 19 days after MRI-PDFF. They were classified into two groups according to weight percentiles: those with obesity (*n* = 53) and those without obesity (*n* = 33). Those with obesity were further classified if they had a BMI > 30 kg/m^2^ (*n* = 13). Based on the results of the MRI-PDFF, 76 patients had steatosis and were divided into four groups according to the PDFF value (S0 to S3). The mean CAP value for S0 showed statistically significant differences in relation to the other groups (S1–S3); however, no significant differences were observed between S1, S2, and S3. A CAP value of 241 dB/m showed good diagnostic performance in distinguishing the presence or absence of steatosis. MRI-PDFF correlated positively with CAP in all participants. However, the ability of CAP to distinguish between different grades of steatosis was limited in participants with a BMI > 30 kg/m^2^, in whom no correlation between CAP and MRI-PDFF values was observed.

A prospective study, conducted by Runge et al. [72] in a European population, compared the accuracy of CAP with ultrasound, using MRS-PDFF as the reference. The study included 62 participants with severe obesity and a mean age of 13.7 years (IQR 12.1–16.1), of whom 60% had steatosis (MRS-PDFF > 4.14%). The CAP values showed significant differences between S0 and the other steatosis grades according to the MRS-PDFF results. However, no significant differences were found when comparing grades S1 versus S2, S1 versus S3, and S2 versus S3. CAP proved to be an effective tool to identify steatosis. Although ultrasonography also showed an acceptable performance, there was no significant difference when compared to CAP (*p* = 0.09). However, the specificity of ultrasonography decreased compared to CAP when only liver parenchymal echogenicity was considered (*p* = 0.04).

Abhinav Anand et al. [73] conducted a prospective cohort study involving 108 Indian adolescents with overweight and obesity, with a mean age of 12.4 ± 1.9 years. They observed a positive correlation between CAP and MRI-PDFF (r = 0.528, *p* < 0.001). The optimal cut-off point for detecting steatosis was 271 dB/m (AUROC 0.745, 95% CI 0.63–0.86). The CAP showed good discriminatory ability in the diagnosis of NAFLD. However, its diagnostic accuracy in differentiating between adjacent grades of steatosis was limited; its ability to distinguish between S0 and S2, S0 and S3, as well as between S1 and S3 was good to excellent.

Lin Yang et al. [74] conducted a study in a pediatric obesity clinic in Asia involving 71 children with obesity. The aim was to assess whether ET could achieve a diagnostic accuracy comparable to MRI-PDFF in predicting steatosis. The cohort was divided into groups according to the grade of hepatic steatosis based on PDFF. Participants with a PDFF >6.4% were classified as NAFLD, while those with a PDFF <6.4% were considered non-NAFLD. CAP was found to be of diagnostic value for NAFLD, especially in the presence of moderate-to-severe steatosis in children with obesity. CAP contributed significantly to the diagnostic value for the S2 and S3 groups, although it did not reach statistical significance in the S1 group.

Finally, Alves et al. [70], using MR elastography, MRI-PDFF, and liver biopsy as references (as previously described), found no significant correlation between CAP and MRI-PDFF (*n* = 16; r = 0.17 (95% CI: −0.34 to 0.61), *p* = 0.5). The authors attributed this result to the small sample size and the lack of a control group.

## 4. Discussion

### 4.1. CAP Performance and Diagnostic Accuracy in Contrast to Liver Biopsy

A limited number of studies have directly compared CAP with liver biopsy [68,69,70]. Although biopsy is considered the gold standard, it is invasive and has inherent limitations due to its nature [40]. For this reason, it is not feasible in apparently healthy individuals and is even less suitable for assessing disease in population-based studies. In this regard, studies that have compared CAP with biopsy have been carried out in a hospital setting, in specialized units, and with a limited number of participants belonging to specific population groups.

Liver biopsy histologically assesses the liver parenchyma obtained from a collected sample, whereas TE analyses a volume of liver tissue that is approximately 100 times larger than that assessed in liver biopsy [75]. Experts recommend that the sample size for biopsy should be at least two to three cm in length and ideally contain at least 11 complete portal tracts [76]. Only Desai et al. [68] describes which criteria were used to consider the biopsy as adequate (≥1.5 cm and a minimum of six portal tracts). Histologically, liver biopsy classifies steatosis according to the percentage of fat infiltration from S0 to S3; S0 (normal) corresponds to <5% of hepatocytes with fat infiltration; S1 (mild) corresponds to between 5 and 33%; S2 (moderate) corresponds to between 33 and 66%; and S3 (severe) corresponds to >66% [30,77]. And although it may be subject to sampling errors, it provides an etiological diagnosis, whereas CAP, although it quantifies liver fat, cannot provide information regarding etiology. However, the optimal CAP cut-off points for diagnosing steatosis in pediatric patients are still under debate. Steatosis is a dynamic process that can change rapidly [36]. In all three studies, biopsies were reviewed retrospectively [68,69,70]. Chaidez et al. [69] reported a shorter mean time between biopsy and CAP assessment. Meanwhile, Desai et al. [68] and Chaidez et al. [69] reported a similar mean CAP value corresponding to S0, with a mean time between tests of 1.3 months and 11 ± 32 days, respectively.

CAP is a non-invasive, harmless, and painless method that is usually acceptable to patients, and the technique is relatively simple to perform. Some children may complain of slight discomfort due to the mechanical impulse generated by the probe in contact with the intercostal space during the test or may even feel a tickle. During the same visit, data related to the presence of fibrosis and steatosis can be obtained, which simplifies the diagnosis of NAFLD. However, in our setting, it is still a method not available in most outpatient clinics and is mainly used at the hospital level in specific clinical research centers or clinics. The quality criteria for CAP are not yet clearly defined. Researchers use the VCTE quality parameters: 10 valid measurements with an IQR < 30% [78,79,80]. Along with this, data such as the fasting hours, probe used, median and interquartile range for elasticity, median and interquartile range for CAP, number of attempted measurements, and number of valid measurements should be recorded. All three studies reported CAP results based on a mean of 10 valid measurements [68,69,70]. However, Alves et al. [70] noted that 34% of examinations were technically unsuccessful (the ratio of valid to invalid measurements was <60%) and that the mean number of attempts needed to obtain 10 valid measurements, in a mostly obese population, was 20, reaching 80 in a single participant with class III obesity. Chaidez et al. [69] admitted that the fasting status data of the 104 subjects (retrospectively recruited) were unknown, while Alves et al. [70] had as one of the exclusion criteria a fasting period of less than 2 h.

There are several types of probes (S, M, and XL), and it is crucial to use the right probe to obtain accurate measurements. In fact, the Fibroscan^®^ automatically suggests the type of probe recommended for the patient during the examination based on SCD. As previously mentioned, SCD includes subcutaneous fat tissue and may attenuate ultrasound signals. Despite the available information on the excellent feasibility and reproducibility of the M and XL probes in the adult population, data comparing the two probes remain contradictory [79,81,82]. In fact, some authors have observed that the optimal cut-off point, when using the XL probe, may be 10 dB/m higher than that of the M probe in patients with NAFLD [83]. Others have pointed out that, if the right probe is used, there are no significant differences [81]. However, it has been observed that the M probe has failure rates of 8% in overweight patients and 17% in obese patients [75]. The XL probe has been developed specifically for overweight or obese patients. However, the S and XL probes have not yet been validated in the pediatric population. All three studies used M and XL probes, and two of them also used the S probe. Desai et al. [68] reported that they excluded from their analysis 67 participants for whom the S probe was used. Chaidez et al. [69] used the S probe based on the participant’s abdominal circumference, while Alves et al. [70] did not use the S probe and excluded participants with chest measurements ≤75 cm. Furthermore, Alves et al. [70] specify that although the use of the XL probe was for research purposes only, as it is not validated for use in children, it was automatically recommended in almost half of the participants under 18 years of age, underlining the importance of obtaining regulatory approval for its use in adolescents under 18 years of age.

CAP is an excellent method for detecting any level of steatosis (S ≥ 1), although its sensitivity is limited for differentiating between histological grades, with data suggesting that it is more superior for detecting steatosis ≥S1 than for steatosis ≥S2 and ≥S3 [84]. However, what is really important is detecting the presence of mild steatosis, for which CAP is better than B-mode ultrasound [79]. Steatosis can eventually lead to steatohepatitis, which in turn can lead to fibrosis; the latter is independently associated with overall long-term mortality, liver transplantation, and liver-related events [85]. Thus, early risk stratification for disease progression is essential for optimal patient management [86]. Desai et al. [68] acknowledged that due to the small sample size, no cut-off points were established to distinguish between mild, moderate, and severe steatosis. Chaidez et al. [69] observed a strong positive correlation between CAP scores and histological steatosis grades, as well as between CAP values and BMI. However, Alves et al. [70] found no correlation between CAP and biopsy. CAP did not differ significantly according to the degree of steatosis (*p* = 0.32).

CAP values can be influenced by various factors, such as patient age, the geographic region of the study, BMI, and the presence of diabetes or NAFLD [51,62,63,84]. All three studies included mostly male participants under the age of 18, and in two of the studies, the percentage of participants by ethnicity was also collected (Desai et al.: 58% white; Chaidez et al.: 53% Hispanic) [68,69]. The percentage of subjects with NAFLD also differed due to the heterogeneity of the study designs. The inclusion criteria of Alves et al. [70] were the suspicion or presumption of NAFLD, while the other two studies included chronic liver disease with clearly different percentages of NAFLD (Desai et al.: 20.3%; Chaidez et al.: 56%) [68,69]. BMI values also differed; Desai et al. [68] obtained a mean BMI within the normal weight range for the entire cohort (22.6 kg/m^2^ (IQR, 19.6–29.2)), but this was higher in subjects with steatosis (28.7 kg/m^2^ (25.4, 36.4)). Chaidez et al. [69] observed a higher mean BMI in the whole cohort (28 ± 8 kg/m^2^), with a higher value in the NAFLD group (33 ± 6 kg/m^2^). And Alves et al. [70] reported a clear predominance of obese subjects in their cohort (94%). Desai et al. [68] found no significant interaction between overweight or obesity and CAP values, although they concluded that the sample size was too small to make any assessment. Chaidez et al. [69] found a positive correlation between BMI and CAP. Alves et al. [70], in a cohort of predominantly obese subjects, observed that technically unsuccessful examinations may also be more frequent in children with obesity, particularly in those with severe obesity.

In summary, compared to liver biopsy, CAP was able to detect the presence or absence of steatosis, as well as to differentiate between certain grades [68]. The three studies observed different optimal cut-off points for identifying steatosis; however, two of them reported a similar mean CAP value for S0 [68,69]. Although it has the ability to predict the histological grade of steatosis [69], some authors did not find a consistent correlation between CAP and the grade of liver steatosis upon biopsy [70]. The ability of CAP to discriminate between different grades of steatosis remains unclear.

### 4.2. CAP Performance and Diagnostic Accuracy in Contrast to Magnetic Resonance

MRI-PDFF has shown a good correlation with liver histology, being able to detect fatty infiltration at levels as low as five percent, making it a superior technique to liver biopsy [51]. This technique accurately identifies the presence of macrovesicular steatosis, predicts histological grade, and even allows for the monitoring of disease progression [51,56,87]. Although MRI-PDFF cannot differentiate between steatosis and steatohepatitis, and liver biopsy remains the only diagnostic tool for assessing and grading the severity of NASH, it is important to note that the pediatric pattern of NAFLD can progress to advanced fibrosis without manifesting the typical features of steatohepatitis [12]. It is a non-invasive technique that offers high levels of accuracy and reproducibility; however, its high cost and limited availability in our practices limit its use at present [51].

The quantification of the percentage of fat in PDFF according to histology is as follows: S0 < 5.5%; S1 5.5–16.2%; S2 16.3–21.6%; and S3 > 21.7% [88]. Unlike biopsy, MRI-PDFF quantifies steatosis of the whole liver, whereas liver biopsy only evaluates a small portion of the liver. It is well tolerated by patients and does not usually require sedation. However, clinical guidelines recommend ruling out other causes of steatosis before diagnosing NAFLD in very young children (<9 years and <3 years) [8,29,30,31].

Similar to studies comparing CAP with biopsy, all studies comparing MRI with CAP have been conducted in hospital settings and specific population groups. These participants are mostly individuals with overweight or obesity and predominantly males. All of them used the M probe, with only two studies reporting the use of the XL probe. The interval time reported between MRI and CAP determination in most studies was relatively short.

As mentioned previously, BMI is a factor to consider when performing measurements with CAP. In fact, the elastography device suggests the probe to use based on the estimated SCD. In participants who are overweight or obese, the thickness of subcutaneous tissue can interfere with the wave propagation in the hepatic parenchyma. In this regard, Shin et al. [71] observed significant differences in mean BMI, abdominal wall thickness (AWT) measured by MRI, and ALT and AST values. M and XL probes were used to perform the CAP measurements. AWT was defined as the thinnest distance between the skin and the liver capsule of the abdominal wall surrounding the liver on an axial image at the level of the main portal vein. AWT was higher in the obese group compared to the non-obese group, and males (2.61 ± 0.63 cm) had significantly higher AWT values than females. They observed a positive correlation between AWT and BMI (r = 0.807, 95% confidence interval (CI): 0.718, 0.870). CAP correlated positively with AWT, whereas MRI-PDFF did not. A significant moderate correlation was found between CAP and MRI, with a high moderate correlation in the non-obese group and a weaker correlation in the obese group, without a significant difference in the correlation between both. In the group of 17 participants with BMI > 30 kg/m^2^, the PDFF value did not correlate with CAP. The authors point out that, although CAP may be a good screening tool to diagnose the presence of steatosis in children, it is probably limited in those with a high BMI, with suboptimal discrimination between steatosis grades 1, 2, and 3 and the CAP value. Attributing this to the sample size, and especially to the low proportion of the S0 group (*n* = 10), would lead to uncertainty in the estimated cut-off value.

In this line, in the study by Abhinav Anand et al. [73] involving a cohort of adolescents with NAFLD and a high prevalence of obesity (84%), CAP demonstrated good discriminatory ability in differentiating between the presence and absence of steatosis and identifying individuals with higher degrees of steatosis. Using the same MRI-PDFF cut-off values for steatosis grade as used by Shin et al. [71] and exclusively using the M probe, they observed a moderate positive correlation between CAP and MRI-PDFF (r = 0.582, *p* < 0.001). In addition, CAP was an independent predictor of MRI-PDFF. In comparison, Shin et al. proposed CAP cut-off values of 241, 299, and 303 dB/m to identify steatosis grades 1–3. Meanwhile, Abhinav Anand et al. [73] observed CAP thresholds for detecting steatosis in children of 271, 296, and 309 dB/m for ≥S1, ≥S2, and S3, respectively. Both cohorts show similar cut-off values, except for a notable difference in the identification of S1. Along the same lines as Abhinav Anand et al. [73], Lin Yang et al. [74] found a statistically significant association between CAP, ALT, and AST in a group of obese adolescents with moderate-to-severe steatosis by MRI-PDFF (*p* < 0.005); however, information regarding the probes used is unknown. In contrast to these results, Alves et al. [70] did not observe a significant correlation between CAP and MRI-PDFF, despite the use of M and XL probes. In addition, ethnic differences between the populations must be taken into account when comparing these studies.

MRS-PDFF is highly accurate in detecting and grading steatosis. Runge et al. [72] reported a 60% prevalence of steatosis based on MRS-PDFF. They point out that what is relevant is detecting the presence of steatosis, rather than the grade of it, as metabolic effects can be seen with fatty infiltration values lower than the currently used cut-off points [88,89]. They defined steatosis grades S1, S2, and S3 as MRS-PDFF fat fraction percentages of >4.14%, >15.72%, and >20.88%, respectively. These cut-off points were validated in a previous study by comparison with liver biopsy. A higher fat content determined by MRS-PDFF was associated with higher CAP values. CAP was able to significantly differentiate between the presence and absence of steatosis (S0 and S1 to S3), but not between the different grades (S1 and S2; S1 and S3; and S2 and S3). A value of 277 dB/m, as the cut-off point, was obtained for identifying steatosis in children with obesity. In addition, although CAP had a higher sensitivity for detecting steatosis compared to ultrasound, the difference was not significant. Similarly, Lin Yang et al. [74] found a steatosis detection rate (S1) by ultrasound of only 64%, and the diagnostic value of CAP for S1 was not significant. However, a study comparing CAP with ultrasound (without using a gold standard) found that, even though both tests have a high specificity, CAP was more sensitive in detecting steatosis [89].

In summary, in specific populations of children with overweight, obesity, and or NAFLD, CAP can distinguish between the presence or absence of steatosis, with a moderate correlation between PDFF and CAP values [71,72]. However, it is not clear whether CAP can reliably differentiate between different grades of steatosis [71,72,73]. The variability in optimal cut-off points among these studies may be related to the sample sizes, the heterogeneity of the studies, multiethnicity, and the prevalence of steatosis.

#### Limitations

One of the main limitations of this review is the scarcity of publications in the literature. This could partly be due to the fact that the search was limited to PubMed and EMBASE, excluding abstracts, doctoral theses and studies related to more technical aspects of VCTE and MRI-PDFF. Additionally, being a relatively new technique and not widely accessible in pediatrics outpatient clinics could also be both a cause and a consequence of the limited available information. Furthermore, we highlight the heterogeneity in study designs, mostly conducted in specific population groups, in hospital settings, and with relatively small sample sizes. These factors have, in some cases, interfered with the interpretation and extrapolation of results. This is justified by the inherent limitations of reference standards, costs, and limited accessibility.

## 5. Conclusions

CAP can be a valuable tool for identifying the presence of steatosis in children. It has the capability to detect steatosis, despite challenges in distinguishing between different grades of hepatic fat infiltration. NAFLD has emerged as a public health concern, especially in an aging population in which its prevalence is rising, while the natural history of the disease remains controversial. Larger studies that include the general population are urgently needed. This would provide more comprehensive information (to validate and establish cut-off points for steatosis and to validate probes) and contribute to simplifying the diagnosis of the disease, leading to its inclusion in clinical guidelines and widespread use in the pediatric and adolescent population.

## Figures and Tables

**Table 1 diagnostics-15-00299-t001:** Characteristics of studies comparing CAP versus biopsy in pediatric population.

References	Country	Population	Study Size (*n*)	Age (yrs)	Biopsy–CAP Interval	Fibroscan Probe	S0 * CAP Cut-Off (dB/m)	Optimal CAP Cut-Off (dB/m)	AUROC (95% CI)	Sensitivity/Specificity (%)	PPV/NPV (%)	Correlation
Desai et al. [68]	USA	Hospital	69	16.0 ± 2.9	1.3 months (IQR 0.5–3.2)	M and XL	198 ± 37	>225	0.93 (0.87–0.99)	87/83	71/93	-
Chaidez et al. [69]	USA	Hospital	206	13.7 ± 3.7	11 ± 32 days (range 0–90)	S, M, and XL	201 ± 42	≥259	0.98 (0.96–0.99)	94/91	97/91	r = 0.84 (*p* < 0.0001)
Alves et al. [70]	USA	Hospital	84	15 ± 3.5	45 days (IQR 34, 83)	M and XL	-	** >249 dB/m	-	-	-	biopsy *n* = 6 r = 0.39 (*p* = 0.44)

Abbreviations: AUROC: Area under receiver operating characteristic curve; CI: Confidence interval; NPV: Negative predictive value; PPV: Positive predictive value. * Steatosis categorized as none (S0, <5%), mild (S1, 5–30%), moderate (S2, 31–60%), or marked (S3, >60%). ** CAP >249 dB/m was considered indicative of steatosis grade ≥1.

**Table 2 diagnostics-15-00299-t002:** Characteristics of studies comparing CAP versus imaging tests in pediatric population.

References	Country	Population	Sample Size (*n*)	Age (yrs)	CAP–MRI Interval	Fibroscan Probe	S0 * CAP Cut-Off dB/m	Optimal CAP Cut-Off (dB/m)	AUROC (95% CI)	Sensitivity/Specificity (%)	PPV and NPV (%)	Correlation
Shin J. et al. [71]	South Korea	Hospital	86	13.1 ± 2.7	2.4 ± 5.0	M and XL	228.4 ± 45.9	** S1-S3 vs. S0 241	0.941 (0.868–0.980)	98.7/80	96.2/87.5	r = 0.486 (*p* < 0.001)
Runge et al. [72]	Netherlands	Pediatric obesity clinic	64	13.7 (IQR 12.1–16.1)	RMS-PDFF 0 (IQR 0–4)	M	253	277	*** S0 vs. S1–3 0.795 (0.671–0.888)	^a^ S0 vs. S1–3 75/75	^a^ S0 vs. S1–3 81.8/67.7	-
Abhinav Anand et al. [73]	New Delhi	Hospital overweight and obesity patients	108	12.4 ± 1.9	Same day	M	≤270	** ≥S1, ≥S2, and S3: 271, 296, and 309, respectively	S0 vs. S1–S3 0.745 (0.630–0.859)	S0 vs. S1–S3 69.9/66.7	S0 vs. S1–S3 92.9/26.3	r = 0.528 (*p* < 0.001)
Lin Yang et al. [74]	China	Pediatric obesity clinic	71	-	-	-	250.91 ± 30.63	^a^ ≥S1, S2, and S3: 265, 297.5 and 288, respectively	≥S1, S2 and S3: 0.752 (0.640–0.864) 0.892 (0.778–1.000) 0.814 (0.706–0.923)	≥S1, S2 and S3 66.7/68.7 75/74.6 76.9/and 75.9	-	-
Alves et al. [70]	USA	Hospital	84	15 ± 3.5	41.5 (IQR 5.25, 74.25)	M and XL	-	^b^ CAP > 249 dB/m	1.0 (0.79–1.0)	100/100	-	r = 0.17 (*p* = 0.5)

Abbreviations: AUROC: Area under receiver operating characteristic curve; CI: Confidence interval; NPV: Negative predictive value; PPV: Positive predictive value. * S0: steatosis categorized as none S0, mild S1, moderate S2, or marked S3. ** PDFF cut-off values used were S1 (6%), S2 (17.5%), and S3 (23.3%). *** Steatosis grades were defined as an MRS-PDFF fat fraction: S1 > 4.14%, S2 > 15.72%, and S3 > 20.88%. ^a^ PDFF cut-off values used were as follows: (S0): PDFF < 6.4%; (S1): 6.4% ≤ PDFF < 17.4%; (S2): 17.4% ≤ PDFF < 22.1%; (S3): PDFF ≥ 22.1%. ^b^ CAP > 249 dB/m was considered indicative of steatosis grade ≥ 1.

## Data Availability

No new data were created or analyzed in this study. Data sharing is not applicable to this article.

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
