# Peer review of "Evaluating Pediatric NAFLD with Controlled Attenuation Parameter: A Comprehensive Narrative Review"

_diagnostics, 2025, doi:10.3390/diagnostics15030299_

Round 1
Reviewer 1 Report (New Reviewer)
Comments and Suggestions for Authors
This manuscript was submitted as an Interesting Images to Diagnostics.
This is a report describing that “Evaluating Pediatric NAFLD with Controlled Attenuation Parameter: A Comprehensive Narrative Review”.
Actually, “There are many tools for NAFLD diagnosis using imaging techniques: ultrasonography (US), computed tomography (CT), and magnetic resonance imaging (MRI). (Line 180)
As authors mentioned “Liver biopsy continues to be the gold standard for diagnosis”. (Line 151)
Readers want to know diagnostic value and relashionship in each US, CT, MRI, CAP, and liver biopsies (real pathological diagnosis).
Table 1 and Table 2 were very complicated. Both of them should be revised as sophisticated form.
Author Response
Reviewer 1
We appreciate and are thankful for your suggestions.
Actually, “There are many tools for NAFLD diagnosis using imaging techniques: ultrasonography (US), computed tomography (CT), and magnetic resonance imaging (MRI). (Line 180)
As authors mentioned “Liver biopsy continues to be the gold standard for diagnosis”. (Line 151)
Readers want to know the diagnostic value and relationship in each US, CT, MRI, PC and liver biopsies (actual pathological diagnosis).
Certainly, we have not dedicated a specific section to comparing the diagnostic value of imaging methods and histology. In response to your insightful comment, we have expanded the information in lines 157-162 and added a reference related to this aspect (Castera L. 2019). However, throughout our review, we highlighted the diagnostic value of ultrasound, MRI, and NAC. We have expanded on this information in section 1.2.2. Diagnostic Imaging, adding a reference (Hernaez R. 2011) in lines 193-196. In addition, we have included a sentence and a reference (Tang A 2013) in lines 211-212. In section 1.2.3 (lines 241-242) we have added an explanatory sentence on CAP and liver biopsy, together with a bibliographic reference. We have not detailed the role of CT, although it has been used since the 1970s, as it is not a routine test in pediatric practices due to radiation exposure.
Table 1 and Table 2 were very complicated. Both must be reviewed in a sophisticated way.
In response to your suggestion, we have revised and improved the tables.
Reviewer 2 Report (New Reviewer)
Comments and Suggestions for Authors
I have reviewed in detail the paper entitled: “Evaluating Pediatric NAFLD with Controlled Attenuation Parameter: A Comprehensive Narrative Review”. In this article, the authors conducted a review of the existing literature on the diagnostic accuracy of CAP in pediatric NAFLD. I comment the following:
I consider it a well-written, and well-structured article, with well-defined subtopics. The introduction has the necessary information to understand the objective of the article. Although there is little information on the subject, the authors make an adequate review of the literature and summarize the studies in a concrete and understandable way.
I consider that the tables are well made; they accurately summarize the articles reviewed
The discussion is extensive and encompasses the points made in the article.
Author Response
Revisor 2
I have reviewed in detail the paper entitled: “Evaluating Pediatric NAFLD with Controlled Attenuation Parameter: A Comprehensive Narrative Review”. In this article, the authors conducted a review of the existing literature on the diagnostic accuracy of CAP in pediatric NAFLD. I comment the following:
I consider it a well-written, and well-structured article, with well-defined subtopics. The introduction has the necessary information to understand the objective of the article. Although there is little information on the subject, the authors make an adequate review of the literature and summarize the studies in a concrete and understandable way.
I consider that the tables are well made; they accurately summarize the articles reviewed
The discussion is extensive and encompasses the points made in the article.
We appreciate your kind comment and hope that this review will be useful for the understanding of an increasingly prevalent disease in this population.
Reviewer 3 Report (New Reviewer)
Comments and Suggestions for Authors
This review article is certainly a good addition to current papers in the field.I have inly a few aspects that should be included in the text.
Minor points:
1. There is a mix of terms of NAFLD and others. So, you better use throughout in the text only the updated terms like MA...
2. I am puzzled describing a rather uniform of the disease. For instance, many drugs can cause a fatty liver, a diagnosis that requires proof by using the updated RUCAM published in 2016.
3. It is common clinical knowledge that patiens of the cohort you are focusing use regular drugs to treat complications associated with the disease.
4. Abundant reports described the development of DILI in these patients, as ascerteined in some reports by the use of the updated RUCAM. You better include the DILI and RUCAM aspects in your text. You may consider work of Patel et al on Drug induced steatohepatitis published 2013, who referenced under #8 and 9 the original RUCAM papers of 1993 but did not quote the updated RUCAM wich was published only 3 years later. Pleasee discuss and quote the paper of Bokan et al of 2020 on drug induced liver injury (DILI)and NAFLD regarding DILI and RUCAM.
5. CAP may be fortified by other clinical parameter such a drug involvement.
Author Response
Revisor 3
We appreciate the reviewer's suggestions.
Minor points:
- There is a mix of terms of NAFLD and others. So, you better use throughout in the text only the updated terms like MA...
We appreciate your insightful observation. We also reflected on the change in nomenclature and which term should be used in our review. As explained in section 2. Materials and Methods (lines 263-265), we decided to use the previous nomenclature, as it was the predominant term used in the reviewed articles. However, we found it appropriate to provide a brief explanation of the change in nomenclature, as outlined in section 1.2. Assessment of Steatosis in Children and Adolescents.
- I am puzzled describing a rather uniform of the disease. For instance, many drugs can cause a fatty liver, a diagnosis that requires proof by using the updated RUCAM published in 2016.
We appreciate your observation. Indeed, other liver diseases can be masked by NAFLD. The change in nomenclature, under the umbrella of steatotic liver disease to liver lipid accumulation disorders, has been well received by pediatric societies. SLD includes several subcategories (MASLD, MASLD overlap, other single etiology, and cryptogenic). In response to your kind suggestion, we have clarified the concept in section 1.2. Assessment of Steatosis in Children and Adolescents (lines 137-138 and 140-144).
Furthermore, we highlight your insightful comment regarding the presence of DILI in this population, which is often attributed to adults with the belief that it is rare in children, despite being a phenomenon also observed in this group. Given the increasing prevalence of SLD, the diagnostic algorithm proposed in the March 2024 consensus document emphasizes the importance that all patients referred with suspected MASLD undergo investigation for a variety of chronic liver disease causes, whether they meet one or more cardiometabolic criteria compatible with MASLD. We have clarified the concept in section 1.2. Assessment of Steatosis in Children and Adolescents (lines 140-144).
- It is common clinical knowledge that patiens of the cohort you are focusing use regular drugs to treat complications associated with the disease.
We appreciate your feedback. However, please note that the studies included in this article did not provide information on the medications used. Their main objective was to evaluate diagnostic methods for the disease. As such, we did not have data on medication use to include in our study.
- Abundant reports described the development of DILI in these patients, as ascerteined in some reports by the use of the updated RUCAM. You better include the DILI and RUCAM aspects in your text. You may consider work of Patel et al on Drug induced steatohepatitis published 2013, who referenced under #8 and 9 the original RUCAM papers of 1993 but did not quote the updated RUCAM wich was published only 3 years later. Pleasee discuss and quote the paper of Bokan et al of 2020 on drug induced liver injury (DILI)and NAFLD regarding DILI and RUCAM.
Thank you for your valuable suggestions regarding the inclusion of DILI and RUCAM aspects in our text. While DILI is indeed a relevant and intriguing topic, it was not the primary focus of our review. However, in the introduction, Section 1.1. Understanding NAFLD in the Pediatric Population, we mention the need to exclude secondary disorders, such as the use of steatogenic drugs, when defining pediatric NAFLD (line 66). In response to your comment, we have reviewed and cited the paper by Bokan et al. (2020) in Section 1.2. Assessment of Steatosis in Children and Adolescents (lines 140-144).
- CAP may be fortified by other clinical parameter such a drug involvement.
As mentioned in the review (lines 495-496), CAP values can be influenced by various factors, such as patient age, geographic region of the study, BMI, and the presence of diabetes or NAFLD. Although drug consumption can influence hepatic fat accumulation, the studies we reviewed do not directly address the impact of medications on the outcome. It would be interesting to explore this approach in future studies or reviews.
Round 2
Reviewer 1 Report (New Reviewer)
Comments and Suggestions for Authors
This paper has been well revised.
This manuscript is a resubmission of an earlier submission. The following is a list of the peer review reports and author responses from that submission.
Round 1
Reviewer 1 Report
Comments and Suggestions for Authors
This is a narrative review article summarizing the usefulness of controlled attenuation parameter (CAP) in diagnosing pediatric patients with non-alcoholic fatty liver disease (NAFLD) compared to the standard tools (liver biopsy and MRI). The authors reviewed articles published between 2016-2023 and included eight articles (three comparing to biopsy, five to MRI), and they concluded that there were the lack of enough evidence and further research was needed. However, the following concerns need to be addressed:
Comments
1. The clinical implication of this review is unclear. CAP itself is an established diagnostic tool for NAFLD in adult patients even compared to liver biopsy and MRI to some extent. There was no description regarding the specific problem of CAP in pediatric patients. If it is a size of fibroscan probe, the authors should summarize the result by the probe size.
2. The number of articles included in this study especially for CAP vs. biopsy (gold standard tool) was so small, and the authors need to investigate articles before 2016.
3. The conclusion part included duplication with the introduction part. The authors should clarify the gap of knowledge in the current clinical practice and state the answer for the gap in the conclusion section.
4. Discussion part included the result of reviewing process and these parts should be separated.
Reviewer 2 Report
Comments and Suggestions for Authors
